# Effects of γ-Irradiation on Structure and Functional Properties of Pea Fiber

**DOI:** 10.3390/foods11101433

**Published:** 2022-05-16

**Authors:** Tianfu Cheng, Caihua Liu, Zhaodong Hu, Zhongjiang Wang, Zengwang Guo

**Affiliations:** College of Food Science, Northeast Agricultural University, Harbin 150030, China; ctf2303@163.com (T.C.); 17686963949@163.com (C.L.); 2021nodanger@163.com (Z.H.); wzjname@126.com (Z.W.)

**Keywords:** γ-irradiation, pea fiber, structural properties, functional properties

## Abstract

In this study, pea residue reserve insoluble diet fiber (hereinafter referred to as pea fiber) was used as a raw material. The effects of γ-irradiation doses (0, 0.5, 1, 2, 3, and 5 kGy) on the structural properties (main composition, particle size and specific surface area, scanning electron microscope (SEM) microstructure, Fourier transform infrared spectroscopy, and X-ray diffraction) and functional properties (oil-holding capacity, swelling and water-holding capacity, and adsorption properties) of pea fiber were explored. The results show that, when the γ-irradiation dose was 2 kGy, compared with the untreated sample, the contents of cellulose, hemicellulose and lignin in pea fiber decreased by 1.34 ± 0.42%, 2.56 ± 0.03% and 2.02 ± 0.05%, respectively, and the volume particle size of pea fiber decreased by 17.43 ± 2.35 μm. The specific surface area increased by 23.70 ± 2.24 m^2^/kg and the crystallinity decreased by 7.65%. Pore and irregular particles appeared on the microstructure surface of the pea fiber treated with γ-irradiation. The results of the infrared spectrum showed that the hemicellulose and lignin in pea fiber were destroyed by γ-irradiation. These results indicate that γ-irradiation can significantly affect the structural properties of pea fiber. When the γ-irradiation dose was 2 kGy, the highest oil-holding capacity, swelling capacity and water-holding capacity of pea fiber were 8.12 ± 0.12 g/g, 19.75 ± 0.37 mL/g and 8.35 ± 0.18 g/g, respectively, and the adsorption capacities of sodium nitre, cholesterol and glucose were also the strongest. These results indicate that the functional properties of pea fiber are improved by γ-irradiation. In this study, γ-irradiation technology was used as pretreatment to provide a theoretical basis for the application of pea fiber in food processing.

## 1. Introduction

Pea is a leguminous plant, which is widely planted all over the world because of its strong adaptability. According to the United Nations, the area of pea harvested in 2018 was behind only soybean, common bean, chickpea and cow pea, while the production of pea ranked fourth, behind soybean, common bean, and chickpea [1]. Pea is one of the most important edible beans in China. In recent years, the pea deep-processing industry has developed rapidly in China. The main products are pea starch, pea protein, pea protein peptides, and pea protein artificial meat. Most of the by-products of pea processing are crushed and used as animal feed, with low added value. Pea dregs are one of the main by-products of pea processing. It is rich in dietary fiber, with a total dietary fiber content of 14–26%, insoluble dietary fiber content of 10–15%, and soluble dietary fiber content of 2–9% [2]. Dietary fiber has the effect of reducing the cholesterol content in blood, preventing heart disease, controlling blood sugar, preventing diabetes, promoting gastrointestinal peristalsis, preventing constipation, and removing harmful toxins from the human body [3]. However, after the unmodified insoluble dietary fiber is directly incorporated into baked food, meat products, beverage products, jams and other fields, it will affect the sensory quality and physical and chemical properties of food and cannot be directly applied in large quantities [4]. As a result, a new process has been employed to improve the application quality and processing characteristics of insoluble dietary fiber in pea.

γ-irradiation is a cold sterilization technology with simple operation, short processing time, large processing capacity, easy control, no chemical requirements and no pollution to the environment. It has broad application prospects in industrial production [5]. Studies show that irradiated food is safe when the dose is less than 10 kGy. The WTO announced that when the irradiation dose is greater than 10 kGy, there is no harm to food [6]. At present, irradiation treatment in the food field is mainly used in food sterilization, protein modification, enzyme elimination, promote fiber hydrolysis and other applications. It was found that γ-irradiation alters the structure of the polymer by forming several intermediates that follow several rapid reaction pathways and form new bonds within the polymer chain [7]. Studies by Fei [8] and Li [9] found an increase in irradiation dose; depolymerization of hemicellulose, cellulose, and lignin; and the association of a cracking structure with a change in the dietary fiber. They also found that, as the free group increased, dietary fiber in the fiber form of crystalline and amorphous forms was destroyed, and more of the crystalline form was broken into an amorphous form. This reduced crystallinity, degree of polymerization, and thermal stability. However, the enzymatic hydrolysis yield of dietary fiber increased. Zhu [10] found that the combination of γ-irradiation and micropulverization has the best degradation effect on soybean dietary fiber, and can improve the physical and chemical properties of soybean dietary fiber. Therefore, it is regarded as an ideal method to improve the quality of soybean dietary fiber. These studies indicate that γ-irradiation treatment is a promising technology and has a prospective application in changing fiber structure and improving fiber function.

Studies have proved that γ-irradiation technology has the effect of changing fiber structure, as well as its physical and chemical properties; it is mainly used in food component modification. However, there are few reports on the effects of irradiation on the physicochemical and functional properties of pea residue dietary fiber. Therefore, in this study, pea residue insoluble dietary fiber (hereinafter referred to as pea fiber) was used as a raw material to explore the effects of γ-irradiation doses (0, 0.5, 1, 2, 3, and 5 kGy) on the physicochemical and functional properties of pea fiber. This study provides a theoretical basis for the wide industrial application of pea fiber in the future.

## 2. Materials and Methods

### 2.1. Materials and Reagents

KBr (Beijing Chemical Reagent Factory, Beijing, China), P-aminobenzene sulfonic acid (Nanjing Chemical Reagent Company, Nanjing, China), Naphthalene hydrochloride (Tianjin Dongli Tianda Chemical Reagent Factory, Tianjin, China), Dinitrosalicylate (DNS), Nitrite and Glucose (Sigma Chemical Company, St. Louis, MO; USA).

### 2.2. Sample Prepare

The irradiation test was carried out at No.2 cobalt source facility of Heilongjiang Institute of Atomic Energy (activity, 200 kCi; temperature, 20 ± 1 °C; average dose rate, 10 Gy/h; inhomogeneity < 5%). The dried pea fiber samples were crushed by ST-G200 high-speed mill (Beijing Xuxinshengke Co., LTD., Beijing, China), and passed through a 60-mesh sieve. The irradiation experiments were carried out after the samples were packed in Ziplock bags (polyethylene). The irradiation doses were 0.5, 1, 2, 3 and 5 kGy, respectively. The irradiation dose of the samples was tracked by a silver dichromate chemical dosimeter (RadEye GF-10, Thermo Fisher Scientific, Waltham, MA, USA), and the absorbed dose was measured by a UV–visible spectrophotometer (SPECORD® 210 PLUS, Analytik Jena AG, Jena, GER). After irradiation, samples were stored in sealed polyethylene bags at room temperature.

### 2.3. Determination of Structural Properties of Pea Fiber

#### 2.3.1. Determination of Main Components

Referring to the research method of Guo et al. [11], cellulose and hemicellulose were prepared by alkali method (24% KOH for 2 h and 10% KOH for 16 h), and lignin was prepared using the acid method (72% H_2_SO_4_ for 1 h). The separated cellulose and lignin were dried at 105 °C for 2 h and hemicellulose was dried at 60 °C for 24 h for further analysis.

#### 2.3.2. Determination of Particle Size and Specific Surface Area

The 3 g sample was dissolved in absolute ethanol, and ultrasonically dispersed for 3 min using an MS2000 Masterizer particle-size analyzer (Malvern company, Malvern, UK) to determine the volume-average particle size and specific surface area of pea fibers.

#### 2.3.3. Determination of Microstructure

The method of determination of microstructure is modified according to Li et al. [12]. The microstructure of pea fiber was determined by SU8010 field emission scanning electron microscope (Hitachi, Japan). The sample particles of about 1 mg were placed on the tape of the circular aluminum sample sub and coated with palladium for 90 s at a current of 15 mA. The specimen stubs were then placed in the observation room. These samples were observed at an accelerator potential of 5 kV with a 6000x increase.

#### 2.3.4. Determination of Fourier Transform Infrared (FTIR) Spectroscopy

According to the method of Guo et al. [13], the pea fiber and KBr (1:250 *w*/*w*) were fully mixed and pressed into a disk, and then Scimitar 2000 FTIR spectrometer (Agilent, Santa Clara, CA, USA) was used to obtain the infrared spectrum of the sample with a wave number of 400–4000 cm^−1^. Each sample was scanned 32 times.

#### 2.3.5. Determination of X-ray Diffraction

The crystal structure determination of pea fiber was slightly modified with reference to the research method of Yang et al. [14]. Scanning from 10 to 40° (2θ°) at 1 π/min was performed with an XPert Powder Multifunctional Powder X-ray diffractometer (Dandong Tonda Technology Co., Ltd., Dandong, China). The generator voltage was 40 kV, and the incident current was 40 mA. The formula of relative crystallinity (RC) of pea fiber is as follows:(1)$$\mathrm{RC}(\%)=\frac{A_{c}}{A_{c}{+A}_{a}}\times 100$$
where A_c_ is the area of the crystalline region; Aa is the area of the amorphous region.

### 2.4. Determination of Functional Properties of Pea Fiber

#### 2.4.1. Determination of Oil-Holding Capacity

Referring to the oil-holding capacity determination method of Liu et al. [15], 0.5 g (M0) pea fiber was mixed with 5 mL soybean oil. After standing in a centrifuge tube for 24 h, the samples were centrifuged for 20 min at 4000 rpm in a refrigerated high-speed centrifuge (X1R, Thermo Fisher Scientific, Waltham, MA, USA). The supernatant was removed, and the weight of the residue was recorded as M1. The oil-holding capacity (OHC) of pea fiber is calculated as follows:(2)$$\mathrm{OHC}\left(g/g\right)=\frac{M_{1}{-M}_{0}}{M_{0}}$$
where, *M_0_* and *M_1_* are the weights of pea dietary fiber before and after oil absorption (g).

#### 2.4.2. Determination of Swelling and Water-Holding Capacity

According to the determination method of Wang et al. [16], the pea fiber suspension was hydrated for 24 h after diluting 0.3 g (*m_0_*) of pea fiber to 15 mL (*V_0_*) with deionized water; the volume was recorded as V1. Then, the pea fiber suspension was centrifuged at 6000 rpm for 20 min. The supernatant was removed and wet fiber weight, m1, was recorded. Swelling (SC) and water-holding capacity (WHC) were calculated using the following formula:(3)$$\mathrm{SC}(\mathrm{mL}/g)=\frac{V_{1}{-V}_{0}}{m_{0}}$$
where *V_0_* is the dilution volume of pea fiber 15 mL; V1 is the volume of the suspension after pea fiber hydration for 24 h (mL).
(4)$$\mathrm{WHC}\left(g/g\right)=\frac{m_{1}{-m}_{0}}{m_{0}}$$
where *m_0_* and *m_1_* are the weight of pea fiber before and after water absorption (g).

#### 2.4.3. Determination of Adsorption Properties

##### Nitrite

According to Luo et al. [17], the adsorption capacity of pea fiber to NaNO_2_ was measured. A 0.1 g sample was added to a 5 mL 20 μg/mL NaNO_2_ solution; the environments of the small intestine and stomach were simulated at pH = 7.0 and pH = 2.0, respectively. Then, the obtained mixture was left to stand at room temperature for 2 h, centrifuged at 4800 rpm for 10 min, and then the 0.5 mL of supernatant was placed in a glass tube. According to Gan et al. [18], NaNO_2_ levels in the supernatant were determined using p-aminobenzene sulfonic acid and naphthalenediamide hydrochloride. Deionized water was added to the tube until the volume of the mixture was 2 mL; then, 2 mL p-aminobenzene sulfonic acid (4 μg/mL) and 1 mL naphthalene hydrochloride (2 μg/mL) were added to the mixture. The solution was left in the dark to react for 30 min, at 538 nm. This enabled us to measure the concentration of NaNO_2_ and obtain the standard curve value (y = 1.9832x + 0.0467, R2 = 0.9981, y is the absorbance value, x is the concentration of NaNO_2_). The adsorption capacity of pea fiber to NaNO_2_ (NIAC) was calculated as follows:(5)$$\mathrm{NIAC}(\mu g/g)=\frac{C_{1}{-C}_{2}}{W}\times V$$
where *C_1_* and *C_2_* are, respectively, the concentration of NaNO_2_ in the supernatant before and after adsorption (μg/L), W is the weight of pea fiber (g), and V is the volume of NaNO_2_ solution (mL).

##### Cholesterol

Cholesterol was added using the method used by Benitez et al. [19] with slight modifications. Fresh egg yolks were diluted with 9 times the weight of distilled water, then beaten until completely emulsified. Next, 0.5 g of pea fiber was added to 25 mL of the egg yolk emulsion and stirred to combine; then, the solution was shaken at 37 °C for 2, 5, 10, 15, 25, 40, 60, 90 and 120 min, respectively. After centrifugation at 4000 rpm for 15 min, the cholesterol content in the supernatant was determined at a wavelength of 550 nm using the phthalaldehyde method and a UV-2700 spectrophotometer (Shimadzu Company, Kyoto, Japan) (Gan et al., 2020). The standard curve was obtained (y = 1.6578x + 0.0254, R2 = 0.9975, y is the absorbance value, x is the cholesterol concentration). The cholesterol adsorption capacity (CAC) of pea fiber was calculated using the following equation:(6)$$\mathrm{CAC}\left(\mathrm{mg}/g\right)=\frac{C_{1}{-C}_{2}}{W}$$
where *C_1_* and *C_2_* are the weight of cholesterol before and after adsorption (mg), and W is the weight of pea fiber (g).

##### Glucose

According to the method of Ma et al. [20] and Chen et al. [21], the glucose adsorption capacity of pea fiber samples was determined. A 0.5 g sample of pea fiber and 100 mL glucose solutions with concentrations of 10, 50, 100 and 200 mmol/L were prepared; they were then shaken at 37 °C for 6 h, centrifuged at 4000 rpm for 15 min, and retained in the supernatant. The glucose content in the supernatant was determined by dinitro salicylate (DNS) chromogenic reagent [18]. Next, 0.5 mL of supernatant was added into a glass tube, wherein deionized water was added until the volume reached 3 mL. This was then mixed with 2 mL dinitrosalicylate (DNS) chromogenic reagent. The mixture was continuously shaken in a water bath at 100 °C for 6 min. After the solution was cooled to room temperature, the glucose concentration was measured at 520 nm to generate a standard curve (y = 14674x + 0.0543, R2 = 0.9983, y is the absorbance value, x is the glucose concentration). The calculation formula of glucose adsorption capacity (GAC) of pea fiber is as follows: (7)$$\mathrm{GAC}(\mathrm{mmol}/g)=\frac{G_{1}{-G}_{2}}{W}\times V$$
where *G_1_* and *G_2_* are glucose concentration before and after adsorption, respectively (mmol/g), W is weight of pea fiber (g), and V is volume of supernatant (mL). 

### 2.5. Statistical Analysis

Three pea fiber samples were prepared, and all samples were tested for three times in parallel. The data were analyzed by one-way ANOVA with SPSS 22.0 software. The data result was mean ± SD, and the difference was significant with *p* < 0.05. Origin 9.0 software was used for data analysis, fitting and standard curve drawing. The areas of crystalline and amorphous regions of X-ray diffraction were calculated by Peakfit version 4.12 software.

## 3. Results and Discussion

### 3.1. Analysis of the Effect of Electron Beam γ-Irradiation on the Physicochemical Properties of Pea Fiber

#### 3.1.1. Analysis of Content of Main Components

According to Table 1, compared with the untreated sample, the contents of cellulose, hemicellulose and lignin in pea fiber significantly decreased (*p* < 0.05) following γ-irradiation. The contents of cellulose, hemicellulose and lignin significantly decreased (*p* < 0.05) with the increase in γ-irradiation dose. This may be a lignin with strong covalent bonds, hydrogen bonds and hemicellulose to form a stable compound composed of cellulose and wrapped up, but it is difficult to predict for certain in outside conditions decomposition [22]. Additionally, γ-irradiation of strong energy rays can promote the depolymerization of hemicellulose and cellulose; lignin association cracking occurred, and the part of dietary fiber component was transformed into oligosaccharides; then, the composition of insoluble dietary fiber in pea dregs changed [23]. The energy absorbed in the irradiation process destroyed the monomeric units of hemicellulose and lignin, cracking into small molecular components or other free radicals, thereby reducing their content [11]. With the increase in γ-irradiation dose, the structure of pea fiber was opened, the exposed cellulose was destroyed and degraded, and the content of pea fiber decreased [24]. The results showed that the binding sites and structures of cellulose, hemicellulose and lignin in pea fiber were destroyed by γ-irradiation, which changed the content of pea fiber.

#### 3.1.2. Analysis of Particle Size and Specific Surface Area

According to Table 2, compared with the untreated sample, with the increase in γ-irradiation dose, the average particle size of pea fiber volume decreased significantly, and the specific surface area increased significantly (*p* < 0.05). Combined with the results in Section 3.1.1, γ-irradiation causes the depolymerization of lignin and hemicellulose in pea fibers by breaking molecular chains, reducing the average particle size and enhancing the specific surface area. Guo [11] and Al-Sheraji et al. [25] also showed that irradiation would destroy the glycosidic bonds between cellulose and reduce cellulose, resulting in a decrease in the particle size of dietary fiber and an increase in the specific surface area of insoluble dietary fiber. This indicates that γ-irradiation can affect the structure and physicochemical properties of pea fibers by breaking glycosidic bonds and changing fiber components.

#### 3.1.3. Analysis of SEM

According to Figure 1, the surface microstructure of untreated samples shows fiber strip structure, and cracks and pores appear in samples treated by γ-irradiation. When the dose of γ-irradiation increased from 0 to 2 kGy, the pore size of the samples gradually increased. When the irradiation dose was 2 kGy, the largest pores appeared in the fiber structure of the sample. When the irradiation dose increased from 2 kGy to 5 kGy, the pore size of the sample showed a downward trend, and when the irradiation dose reached 5 kGy, the surface of pea fiber breaks and forms a lamellar microstructure. This is consistent with the specific surface area results in Section 3.1.2. This may be due to the fact that the increase in γ-irradiation intensity can break the glycosidic bond in the molecular chain, reduce the molecular weight and weaken the interaction between molecules, promoting the formation of carbonyl and double bonds, and leading to the relaxation of the pea fiber’s structure, which demonstrated a honeycomb structure and had larger pores [26,27]. The decrease in particle size and changes in the microstructure of pea fibers may generate capillary action and form a larger specific surface area, which may be important for its absorption capacities with some other compounds [17]. Jiang et al. [28] also demonstrated that high-intensity energy field treatment can change the surface structure and affect the specific surface area by destroying the intermolecular crosslinking of fibers. When the irradiation dose is too high, layered substances appear on the surface of pea fiber. This may be due to the accumulation of residual protein on the surface of the fiber and degraded fiber fragments caused by excessive dose irradiation. The research shows that the hydration properties and glucose absorption capacity of dietary fiber are mainly related to the porosity of the fiber structure [29,30]. This indicates that the appropriate dose γ-irradiation treatment can improve the functional activity by adjusting the microstructure and porosity of pea fiber.

#### 3.1.4. Analysis of Fourier Transform Infrared Spectroscopy

Fourier transform infrared spectroscopy is an infrared absorption spectrum formed according to the vibration of molecules at different wavelengths, which is used to detect changes in molecular groups and chemical bonds of pea fibers. The position and strength of the absorption peak are mainly affected by the types of chemical bonds or molecular groups, and the position of the absorption peak changes with induction, conjugation or steric hindrance [31]. According to Figure 2, all the pea fiber samples showed similar spectral curves while retaining characteristic bands specific to each procedure. The absorption bands of all pea fibers in the range of 3000–3700 cm^−1^ are due to O-H bond tensile vibration, and these absorption bands also indicate the presence of pectin and hemicellulose in soybean residue fibers [32]. The absorption peaks at 2853 cm^−1^ and 2925 cm^−1^ were attributed to the asymmetric and symmetrical C-H vibrational bands in the polysaccharide compound methylene [33]. The absorption peak was observed at 1235 cm^−1^, indicating the presence of a crystalline region. With the increase in γ-irradiation dose, the intensity of these absorption bands weakened, probably due to the destruction of intramolecular hydrogen bonds in cellulose and hemicellulose compared with untreated samples [34]. The intensity of the absorption peak at 1725 cm^−1^ weakened, indicating that the adsorption of water on the fiber matrix became weak [35]. The decrease in absorption bands near 1000 cm^−1^ was due to the stretching of C=O and aromatic skeleton of aldehyde/ester groups of hemicellulose and lignin, indicating that the hemicellulose and lignin in pea fiber were destroyed by γ-irradiation [36]. The reactive groups play an important role in the physicochemical and functional properties of dietary fibers, such as hydration, adsorption, cation exchange capacity, and metal chelation [37]. This indicates that γ-irradiation treatment can modulate physicochemical and functional properties by altering the reactive groups of pea fibers.

#### 3.1.5. Analysis of X-ray Diffraction

The effect of γ-irradiation on the crystalline properties of pea fiber was studied by X-ray diffraction. The overall peak shape represents the crystal type of the sample. The increase in diffraction peak intensity indicates that the crystallinity at the diffraction angle increases. It can be seen from Figure 3 that the irradiation treatment does not cause the peak shape to change, and each sample has obvious absorption peaks at the scanning angles (2θ) of 19.96° and 34.56° (except 2 kGy, which is 35.05°). This indicates that the crystals of the six groups of fiber samples are all of the type I cellulose type, which consists of ordered crystalline cellulose regions and disordered cellulose and hemicellulose regions, and irradiation treatment does not change the cellulose type of the samples. According to Figure 2 and Figure 3, the highest crystallinity of untreated sample is 34.22%. With the increase in irradiation dose, the crystallinity of pea fiber decreased first and then increased. When the irradiation dose was 2 kGy, the crystallinity was the lowest with a value of 26.57%, which might be because the crystalline form and the amorphous form of pea fiber were destroyed by γ-irradiation, and the destroyed crystalline form was decomposed into the amorphous form, thus reducing the crystallinity of pea fiber [9]. When the irradiation dose was greater than 2 kGy, the γ-irradiation destroyed the monomer units of hemicellulose and lignin in the amorphous region, but the structural damage degree of cellulose monomer units in the crystallization region is small, so the destruction degree of the amorphous region is greater than that of the crystallization region, resulting in the increase in crystallinity [38]. The results showed that γ-irradiation changed the crystal structure of pea fibers by changing the contents of cellulose, hemicellulose and lignin in pea fibers.

### 3.2. Analysis of the Effect of γ-Irradiation on the Functional Properties of Pea Fiber

#### 3.2.1. Analysis of Oil-Holding Capacity

The oil-holding capacity of dietary fiber can significantly improve the sensory characteristics of food, and help to prolong the shelf life of food [12]. Meanwhile, the high oil-holding capacity of dietary fiber can reduce the absorption of lipids in the intestinal tract [39]. According to Figure 4, the oil-holding capacity of pea fiber increased first and then decreased with the increase in γ-irradiation dose. When the γ-irradiation dose was 2 kGy, the oil-holding capacity of pea fiber was the highest. This may be because the γ-irradiation treatment can increase the porosity and specific surface area of pea fibers by destroying the microstructure. This is helpful for powering more oil droplets to contact and embed in pea fibers, thereby improving the oil-holding capacity of pea dietary fibers [25]. However, when the irradiation dose was greater than 2 kGy, the pores on the microstructure surface of pea fiber decreased, and the specific surface area decreased; therefore, that the oil-holding capacity decreased.

#### 3.2.2. Analysis of Swelling and Water-Holding Capacity

The water swelling capacity is a functional characteristic in food processing, which is used to evaluate the hydration properties of dietary fiber [12]. The swelling properties and water-holding capacity of dietary fiber are beneficial to intestinal function by increasing chyme bulk and enhancing peristalsis. According to Figure 5, compared with the untreated sample, the swelling capacity and water-holding capacity of pea fiber were significantly increased (*p* < 0.05). With the increase in γ-irradiation dose, the swelling capacity and water-holding capacity increased first and then decreased. Additionally, when the irradiation dose was 2 kGy, the swelling capacity and water-holding capacity were highest. It shows that the appropriate irradiation dose can significantly improve the swelling capacity and water-holding capacity of dietary fiber. Alam reported in his study that particle size can affect the swelling capacity of dietary fiber, and the smaller the particle size, the higher the swelling capacity of dietary fiber [40]. This is consistent with the results in Section 3.1.2. Furthermore, the moderate irradiation could break some fibers, thereby loosening the dense network structure of dietary fibers and increasing the specific surface area. This is also one of the reasons for the increased swelling capacity of dietary fiber. Irradiation treatment can break the glycosidic bond of dietary fiber, and the microstructure of fiber has obvious pores and honeycomb structure at 2 kGy. This provides more space for the storage of water molecules, thereby enhancing the water-holding capacity of the pea fibers [41,42]. However, when the irradiation dose was too high, the swelling and water-holding capacity of pea fiber decreased significantly (*p* < 0.05). This may be because the swelling capacity is related to the crystallinity. Studies have shown that the crystallinity of γ-irradiated potato starch is reduced, which leads to a reduction in its swelling capacity [43]. At the same time, high-dose irradiation will lead to the destruction of the network structure of the fibers, resulting in a decrease in water-holding capacity [12].

#### 3.2.3. Analysis of Adsorption Properties

##### Nitrite

Nitrite is a relatively toxic compound and can form carcinogenic compounds during human digestion [28]. Some studies have shown that nitro compounds with carcinogenicity to animals can also enter the fetus through the placenta and have teratogenic effects on the fetus [44,45]. As an important indicator, pH has a great influence on the ability of dietary fiber to adsorb nitrite ions. Therefore, the adsorption capacity of each pea fiber sample for nitrite ions under simulated gastric environment (pH = 2) and intestinal environment (pH = 7). Figure 6 shows that the nitrite adsorption capacity of pea fiber is affected by pH value and γ-irradiation dose. When pH = 2, the nitrite adsorption capacity was much higher than that of pH = 7. This may be because, in acidic conditions, NO_2_^−^ reacts with H^+^ to produce HNO2 and then forms nitrogen oxides, including the strong electron affinity compound N_2_O_3_, which can combine with the negatively charged oxygen atoms of phenolic acid groups in dietary fiber and cause adsorption [46]. Another explanation is that the active groups such as uronic acid, amino acid, and especially phenolic acid contained in the structure of dietary fiber have a strong adsorption effect on nitroso groups under acidic conditions; this is why the nitrite ion adsorption capacity of pea fiber sample at pH = 2 is significantly higher than that at pH = 7 [47]. With the increase in irradiation dose, the nitrite adsorption capacity of pea fiber increases first and then decreases at two pH values. When the irradiation dose is 2 kGy, the nitrite adsorption capacity of pea fiber was the highest. The γ-irradiation changed the structure of pea fibers, the surface appeared loose and porous, and the specific surface area became larger, which exposed more adsorption sites for nitrite ions to a certain extent. The capillary effect formed by these pores accelerates the entry of nitrite ions into the interior of the fiber structure, so the adsorption capacity of pea fibers for nitrite ions after moderate irradiation treatment is improved in the gastrointestinal environment.

##### Cholesterol

The occurrence of coronary heart disease is directly related to the content of cholesterol in the blood. There is evidence that dietary fibers reduce the risk of cardiovascular disease and serum cholesterol levels through the absorption of cholesterol [48,49]. Thus, it is of great significance to improve the cholesterol adsorption capacity of dietary fiber. According to Figure 7, the cholesterol adsorption capacity of pea fiber was affected by pH value and γ-irradiation dose. With the increase in γ-irradiation dose, the cholesterol adsorption capacity of pea fiber increased first and then decreased at both pH values. When the γ-irradiation dose was 2 kGy, the cholesterol adsorption capacity of pH = 2 and pH = 7 was the largest, and the cholesterol adsorption capacity of pH = 7 was greater than that of pH = 2. This may be due to the repulsion between H+ and some positive charges in dietary fiber and cholesterol under acidic conditions, which affects the binding of dietary fiber and cholesterol, thus reducing the adsorption amount of cholesterol so that the absorption of cholesterol by pea fiber in intestinal environment is more effective than that in stomach environment. When the irradiation dose was lower than 2 kGy, the γ-irradiation increases the specific surface area of the insoluble dietary fiber of pea dregs and changes the structure of hemicellulose and lignin to enhance the capillary action, thereby increasing the adsorption capacity of cholesterol [18,47,50]. However, when the irradiation dose is too high, the surface pores and surface area of pea fiber decrease, which weakens the effect of capillaries, thus weakening the adsorption effect of pea fiber on cholesterol and reducing the adsorption amount of cholesterol [51].

##### Glucose

The absorption of glucose by dietary fiber can delay or reduce the digestion and absorption of glucose in the gastrointestinal tract, thus playing a role in reducing blood glucose, which is also an important functional characteristic of dietary fiber [12]. It can be seen from Figure 8 that the adsorption capacity of pea fiber on glucose is affected by glucose concentration and γ-irradiation dose. With the increase in γ-irradiation dose, the adsorption capacity of pea fiber to different concentrations of glucose first increased and then decreased. When the dose of γ-irradiation was 2 kGy, the adsorption capacity of pea fibers to different concentrations of glucose was the highest, which indicated that γ-irradiation could improve the glucose adsorption capacity of pea fibers, possibly due to the increase in specific surface area and porosity caused by moderate irradiation, making it easier for glucose to enter the interior of the fiber and bind more tightly to the interior of the dietary fiber. Studies have shown that the increased hydration of dietary fiber makes it easier for glucose to bond to the fiber’s surface [52]; this is confirmed by the findings in Section 3.2.2 of this paper. Furthermore, the adsorption capacity positively correlated with glucose concentration. This may be due to the increased contact probability between glucose and fiber, which improves the adsorption capacity of pea fiber. However, when the irradiation dose was too high, the damage to polar and non-polar groups in pea fiber structure weakens the interaction with glucose molecules [53], thus reducing the adsorption amount of glucose.

## 4. Conclusions

Pea fiber was irradiated with different doses (0, 0.5, 1, 2, 3, and 5 kGy) by γ-irradiation technology to investigate the effects of irradiation dose on the structure and functional characteristics of pea fiber. According to the structural characteristics of pea fiber, when the γ-irradiation dose was 2 kGy, the contents of cellulose, hemicellulose and lignin in pea fiber decreased by 1.34 ± 0.42%, 2.56 ± 0.03% and 2.02 ± 0.05%, respectively, and the crystallinity of pea fiber decreased by 7.65%. The pore and irregular particles appeared on the microstructure surface of pea fiber treated by γ-irradiation. The results of infrared spectroscopy showed that the hemicellulose and lignin in pea fiber were destroyed by γ-irradiation. The results of the functional characteristics of pea fiber showed that, when the γ-irradiation dose was 2 kGy, the highest oil-holding capacity, swelling capacity and water-holding capacity of pea fiber were 8.12 ± 0.12 g/g, 19.75 ± 0.37 mL/g and 8.35 ± 0.18 g/g, respectively. Additionally, the adsorption capacity of sodium nitre, cholesterol and glucose were also the strongest in these conditions. These results suggest that the functional properties of pea fiber can be improved by γ-irradiation, changing the structural properties of pea fiber. In this study, γ-irradiation technology was used as pretreatment to provide a theoretical basis for the application of pea fiber in food processing.

## Figures and Tables

**Figure 1 foods-11-01433-f001:**
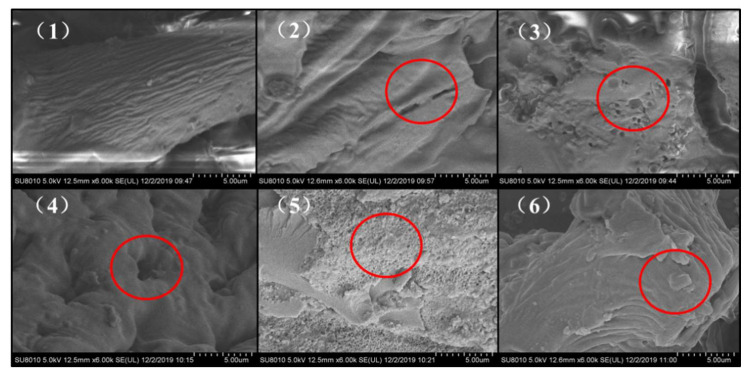
Scanning electron microscope (SEM) images of pea fiber after γ-irradiation. Note: (**1**) as the control, the irradiation doses of (**2**)–(**6**) were 0.5, 1, 2, 3, and 5 kGy, respectively, and the magnification was 6000×. Note: The red circles represent cracks, pores, and layered structures present in the γ-irradiation treatment samples.

**Figure 2 foods-11-01433-f002:**
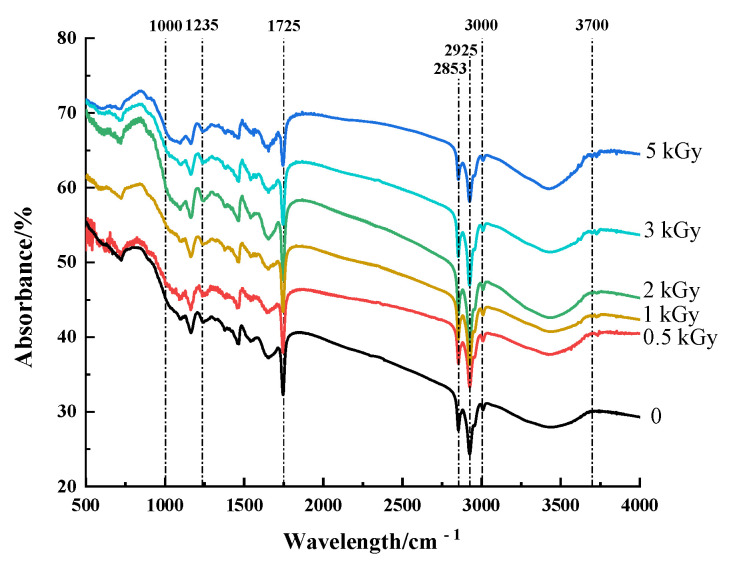
Effect of γ-irradiation on infrared spectrum of pea fiber.

**Figure 3 foods-11-01433-f003:**
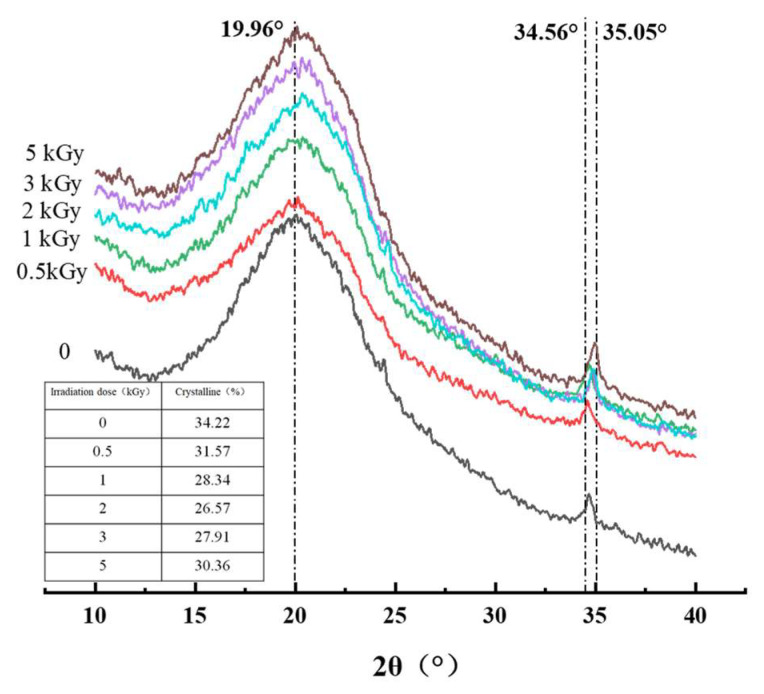
Effects of γ-irradiation on X-ray diffraction of pea fiber.

**Figure 4 foods-11-01433-f004:**
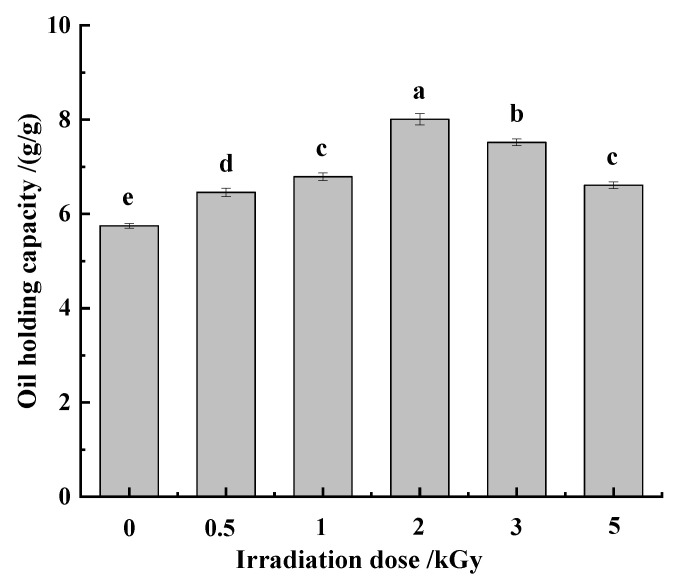
Effect of γ-irradiation on oil-holding capacity of pea fiber. Note: Different lowercase letters indicate significant differences in oil-holding capacity (*p* < 0.05).

**Figure 5 foods-11-01433-f005:**
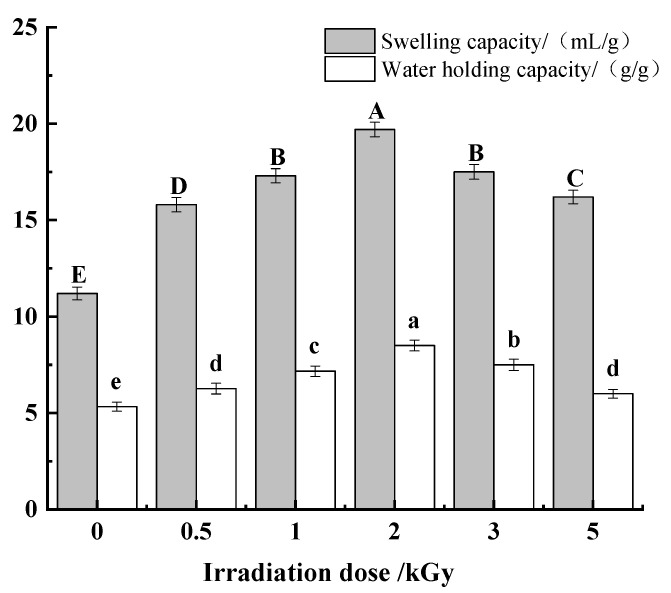
Effects of γ-irradiation on the swelling and water-holding capacity of pea fiber. Note: Different uppercase letters indicate significant differences in swelling capacity (*p* < 0.05), and different lowercase letters indicate significant differences in water-holding capacity (*p* < 0.05).

**Figure 6 foods-11-01433-f006:**
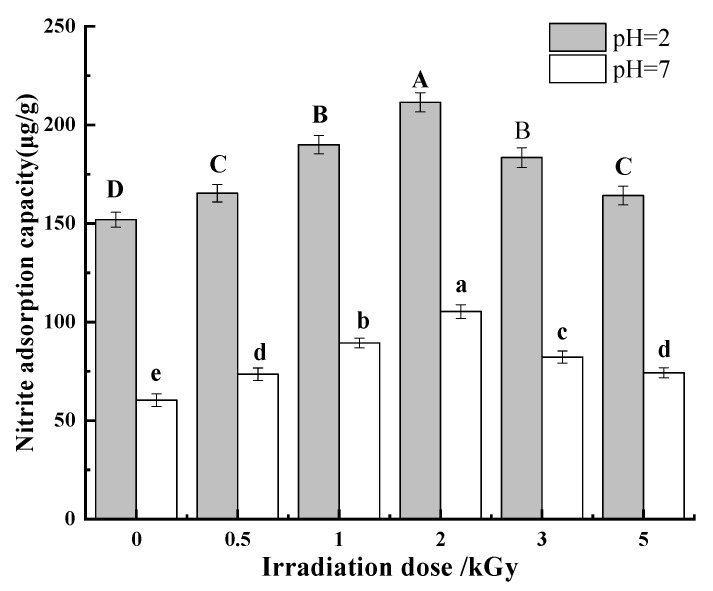
Effect of γ-irradiation on nitrite adsorption of pea fiber. Note: Different uppercase letters indicate significant difference in nitrite adsorption capacity under simulated gastric environment (pH = 2) (*p* < 0.05), and different lowercase letters indicate significant difference in nitrite adsorption capacity under simulated intestinal environment (pH = 7) (*p* < 0.05).

**Figure 7 foods-11-01433-f007:**
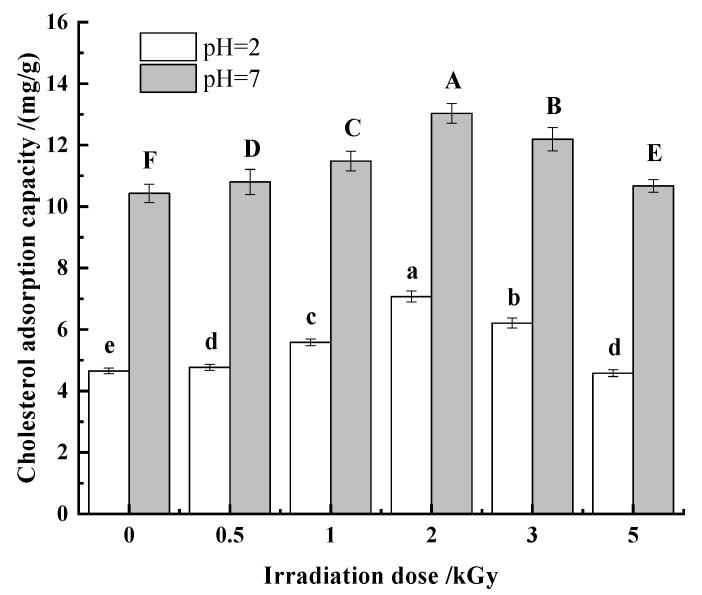
Effect of γ-irradiation on cholesterol adsorption of pea fiber. Note: Different uppercase letters indicate significant difference in cholesterol adsorption capacity under simulated gastric environment (pH = 2) (*p* < 0.05), and different lowercase letters indicate significant difference in cholesterol adsorption capacity under simulated intestinal environment (pH = 7) (*p* < 0.05).

**Figure 8 foods-11-01433-f008:**
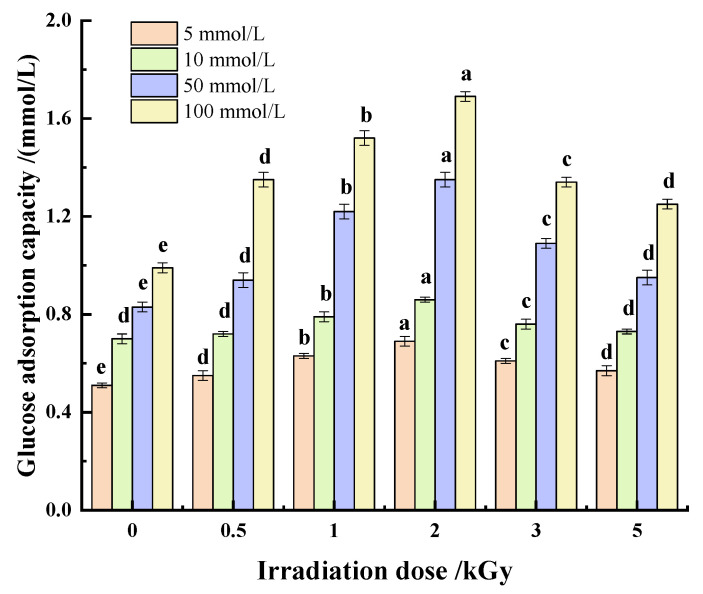
Effect of γ-irradiation on glucose adsorption of pea fiber. Note: Different small letters at the same concentration indicate significant differences in glucose adsorption capacity (*p* < 0.05).

**Table 1 foods-11-01433-t001:** Effects of γ-irradiation on the content of main components of pea fiber.

Treatment Dose (kGy)	Cellulose (%)	Hemicellulose (%)	Lignin (%)
0	37.44 ± 0.18 a	15.10 ± 0.04 a	14.27 ± 0.05 a
0.5	37.09 ± 0.23 b	13.31 ± 0.05 b	13.81 ± 0.06 b
1	36.23 ± 0.15 c	12.97 ± 0.06 c	12.52 ± 0.07 c
2	36.10 ± 0.23 d	12.54 ± 0.07 d	12.25 ± 0.05 d
3	36.25 ± 0.18 c	12.10 ± 0.06 e	12.11 ± 0.06 e
5	36.31 ± 0.23 d	11.50 ± 0.06 f	12.02 ± 0.06 f

Note: Different lowercase letters in the same column indicate significant difference (*p* < 0.05)—same below.

**Table 2 foods-11-01433-t002:** Effects of γ-irradiation on particle size and specific surface area of okra pea fiber.

Treatment Dose (kGy)	D10 (μm)	D50 (μm)	D90 (μm)	Volumetric Mean Particle Size (µm)	Specific Surface Area (m^2^/kg)
0	46.15 ± 2.03 a	75.57 ± 2.33 a	165.47 ± 3.01 a	135.62 ± 3.14 a	78.68 ± 1.42 d
0.5	42.35 ± 2.43 a	73.67 ± 2.06 ab	163.98 ± 3.14 a	133.82 ± 2.61 a	85.36 ± 1.77 c
1	39.98 ± 1.50 b	70.79 ± 1.99 c	155.29 ± 2.67 b	129.38 ± 2.81 ab	94.78 ± 1.91 ab
2	36.16 ± 1.29 c	65.47 ± 2.30 d	153.09 ± 2.93 bc	118.19 ± 2.01 c	102.38 ± 2.54 a
3	37.26 ± 1.43 c	67.19 ± 2.19 cd	154.48 ± 3.01 b	124.29 ± 2.14 b	100.94 ± 2.63 a
5	41.23 ± 1.89 b	70.65 ± 2.31 c	154.98 ± 2.60 b	127.21 ± 1.54 ab	95.79 ± 1.95 b

## Data Availability

Data is contained within the article.

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
