# Peer review of "Effects of γ-Irradiation on Structure and Functional Properties of Pea Fiber"

_foods, 2022, doi:10.3390/foods11101433_

Round 1

Reviewer 1 Report

The reviewed article is based on an interesting scientific idea and the research is carried out competently. The methods are properly selected and the effects of pea fiber modification are described quite comprehensively. However, the article requires improvement in terms of editing. Errors are too numerous to list: missing spaces, surname lower case (fourier), compound names capitalized without need, "PH", no subscripts in chemical formulas, "γ-irradiation" instead of "γ-Irradiation "at the beginning of the sentence, unnecessary constructs (N-nitro) and others. Please, the authors carefully read and correct the entire text in this regard.

Author Response

Dear Editors and Reviewers:

Thank you for your letter and for the reviewers’ comments concerning our manuscript entitled “Effects of γ- irradiation on structure and functional properties of Pea fiber” (ID: foods-1680612). Those comments are all valuable and very helpful for revising and improving our paper, as well as the important guiding significance to our researches. We have studied comments carefully and have made correction which we hope meet with approval. Revised portion are marked in red in the paper. We hope that the revision is acceptable and look forward to hearing from you soon. The main corrections in the paper and the responds to the reviewer’s comments are as following:

Reviewer #1:

- The reviewed article is based on an interesting scientific idea and the research is carried out competently. The methods are properly selected and the effects of pea fiber modification are described quite comprehensively. However, the article requires improvement in terms of editing. Errors are too numerous to list: missing spaces, surname lower case (fourier), compound names capitalized without need, "PH", no subscripts in chemical formulas, "γ-irradiation" instead of "γ-Irradiation "at the beginning of the sentence, unnecessary constructs (N-nitro) and others. Please, the authors carefully read and correct the entire text in this regard.

Response: Thanks for your valuable suggestions. We have made polishing changes to the language of the manuscript. We checked all formats including references, technical terms and chemical formulas. We've fixed a number of editorial issues in the article. And we have modified the text and marked in red.

Reviewer 2 Report

The paper entitled “Effects of γ- irradiation on structure and functional properties 2 of Pea fiber” from Chen  al is not appropriate for publication in Foods. The aim of the research is to study the effects of  γ -irradiation dose (0, 0.5, 1, 2, 3, 5 kGy) on the structural and functional properties (oil holding capacity, swelling and water holding capacity, adsorption properties) of pea fiber. The research reported here could be more a technical report than a research paper. The main concern with this article is the low originality of the topic. Moreover, in my opinion it is not relevant or interesting enough.

Author Response

Dear Editors and Reviewers:

Thank you for your letter and for the reviewers’ comments concerning our manuscript entitled “Effects of γ- irradiation on structure and functional properties of Pea fiber” (ID: foods-1680612). Those comments are all valuable and very helpful for revising and improving our paper, as well as the important guiding significance to our researches. We have studied comments carefully and have made correction which we hope meet with approval. Revised portion are marked in red in the paper. We hope that the revision is acceptable and look forward to hearing from you soon. The main corrections in the paper and the responds to the reviewer’s comments are as following:

Reviewer #2:

- The paper entitled “Effects of γ- irradiation on structure and functional properties 2 of Pea fiber” from Chen  al is not appropriate for publication in Foods. The aim of the research is to study the effects of  γ -irradiation dose (0, 0.5, 1, 2, 3, 5 kGy) on the structural and functional properties (oil holding capacity, swelling and water holding capacity, adsorption properties) of pea fiber. The research reported here could be more a technical report than a research paper. The main concern with this article is the low originality of the topic. Moreover, in my opinion it is not relevant or interesting enough.

Response: Thanks for your valuable suggestions. We have added research purpose and significance in the Introduction section. All the methods in this paper are described in detail. The underlying mechanism of the effect of γ-irradiation on the structural and functional properties of pea fiber has also been re-evaluated in the Results and Discussion section and a citation was added. 

For the structural data, we have marked the peak results in the FTIR figure and XRD figure, and elucidated the underlying mechanism and cited references in the results and analysis. Among them, the analysis results of Fourier transform infrared spectroscopy analyzed the changes of functional groups and main chemical bonds through the changes of peaks. The changes of γ-radiation on pea fibers crystallinity was analyzed by X-ray diffraction.

For data on functional characterization, In the analysis of oil-holding capacity, we clarified the reason why the oil-holding capacity of pea fiber was the highest when the γ-irradiation dose was 2 kGy. This may be because γ-irradiation treatment can increase the porosity and specific surface area of pea fibers by destroying their microstructure. This helps to allow more oil droplets to contact and embed the pea fiber, thereby improving the oil-holding capacity of the pea dietary fiber (Al Sheraji et al., 2012). 

And in the analysis of swelling and water retention, it is elaborated that the appropriate irradiation dose can significantly improve the swelling and water holding capacity of dietary fiber. Alam who reported in his study that particle size can affect the swelling capacity of dietary fiber, and the smaller the particle size, the higher the swelling capacity of dietary fiber (Alam et al., 2014). Besides, the moderate irradiation could break some fibers, thereby loosening the dense network structure of dietary fibers and increasing the specific surface area. This is also one of the reasons for the increased swelling capacity of dietary fiber. Irradiation treatment can break the glycosidic bond of dietary fiber, and the microstructure of fiber has obvious pores and honeycomb structure at 2 kGy. This provides more space for the storage of water molecules, thereby enhancing the water-holding capacity of the pea fibers(Xia et al., 2018; Fan et al., 2020).

When analyzing the nitrite adsorption of pea fiber, we elaborated the adsorption capacity of pea fiber samples for nitrite ions under simulated gastric environment (pH=2) and intestinal environment (pH=7). When pH=2, the nitrite adsorption capacity was much higher than that of pH=7. This may be because under acidic conditions, NO2- reacts with H+ to produce HNO2 and then forms nitrogen oxides, including the strong electron affinity compound N2O3, which can combine with the negatively charged oxygen atoms of phenolic acid groups in dietary fiber and cause adsorption (Lyu et al., 2021). Another explanation is that the active groups such as uronic acid, amino acid, especially phenolic acid contained in the structure of dietary fiber have a strong adsorption effect on nitroso groups under acidic conditions, that is why the nitrite ion adsorption capacity of pea fiber sample at pH = 2 is significantly higher than that at pH = 7 (Chu et al., 2019). With the increase of irradiation dose, the nitrite adsorption capacity of pea fiber increases first and then decreases at two pH values. When the irradiation dose is 2 kGy, the nitrite adsorption capacity of pea fiber was the highest. The γ-irradiation changed the structure of pea fibers, the surface appeared loose and porous, and the specific surface area became larger, which exposed more adsorption sites for nitrite ions to a certain extent. The capillary effect formed by these pores accelerates the entry of nitrite ions into the interior of the fiber structure, so the adsorption capacity of pea fibers for nitrite ions after moderate irradiation treatment is improved in the gastrointestinal environment.

In the analysis of cholesterol adsorption of pea fiber, we added that the γ-irradiation increases the specific surface area of the insoluble dietary fiber of pea dregs at irradiation doses below 2 kGy, and changes the structure of hemicellulose and lignin to enhance the capillary action, thereby increasing the adsorption capacity of cholesterol (Chen et al., 2014; Chu et al., 2019; Gan et al., 2020).

When analyzing the glucose adsorption of pea fiber, we describe the underlying mechanisms. The γ-irradiation could improve the glucose adsorption capacity of pea fibers, possibly due to the increase of specific surface area and porosity caused by moderate irradiation, making it easier for glucose to enter the interior of the fiber and bind more tightly to the interior of the dietary fiber. Studies have shown that the increased hydration of dietary fiber makes it easier for glucose to bond to the fiber's surface (Zhang et al., 2015). Furthermore, the adsorption capacity was positively correlated with glucose concentration. This may be due to the increased contact probability between glucose and fiber, which improves the adsorption capacity of pea fiber.

We have modified the text and marked in red. 

Reviewer 3 Report

Indeed, the topic is very interesting and hot scientifically.. Frankly speeking, some of my studnets are doing a related experiments, and its good to reveiw this manuscripit. 

Here are my points, which I look forward from the authors to fix all before going to the next step::

  • mention in detailes why you select the pea to be the source of fiber, what are the advantges?
  • why you did not use 10 KG of irradiation, is a relation between the dose and the needed effect on the fiber?
  • there are many uses of jargon words, try to limit it.  
  • Methods are the heart of the article,,,, try to detail every thing you used.. for example, in detailes explain how did you irradiate the fiber, fiber extraction, characterization and so on.
  • why you did not identify the fibers structure?
  • Most importantly, the underlying mechanism of the effect of the irradiation on the functional properties of fibers is REALLY missing, try to elucidate it, otherwise...
  • The FTIR figure showed nothing, peak fit these results or provide much better one!! The same with XRD!

Author Response

Dear Editors and Reviewers:

Thank you for your letter and for the reviewers’ comments concerning our manuscript entitled “Effects of γ- irradiation on structure and functional properties of Pea fiber” (ID: foods-1680612). Those comments are all valuable and very helpful for revising and improving our paper, as well as the important guiding significance to our researches. We have studied comments carefully and have made correction which we hope meet with approval. Revised portion are marked in red in the paper. We hope that the revision is acceptable and look forward to hearing from you soon. The main corrections in the paper and the responds to the reviewer’s comments are as following:

Reviewer #3:

- Indeed, the topic is very interesting and hot scientifically. Frankly speeking, some of my studnets are doing a related experiments, and its good to reveiw this manuscripit. Here are my points, which I look forward from the authors to fix all before going to the next step:

  1. mention in detailes why you select the pea to be the source of fiber, what are the advantges?

Response: Thanks for your valuable suggestions. The reasons for choosing peas as a fiber source and its advantages are explained in the Introduction section. Such as:

In recent years, the pea deep processing industry has developed rapidly in China. The main products are pea starch, pea protein, pea protein peptides, and pea protein artificial meat. Most of the by-products of pea processing are crushed and used as animal feed, with low added value. Pea dregs are one of the main by-products of pea processing.  It is rich in dietary fiber, with the total dietary fiber content of 14 % ~ 26 %, insoluble dietary fiber content of 10 % ~ 15 %. However, after the unmodified insoluble dietary fiber is directly incorporated into baked food, meat products, beverage products, jams and other fields, it will affect the sensory quality and physical and chemical properties of food and can't be directly applied in large quantities (Lee et al., 2005). As a result, a new process employed to improve the application quality and processing characteristics of pea insoluble dietary fiber.

  1. why you did not use 10 KG of irradiation, is a relation between the dose and the needed effect on the fiber?

Response: Thanks for your valuable suggestions. Yes. We found the relationship between the γ-irradiation dose and the changes of pea fibers in the preliminary experiments. When the dose is 2 kGy, the γ-radiation can improve the performance of pea fiber better. But when the irradiation dose exceeded 5 kGy, the properties of the pea fibers decreased sharply. Therefore, we chose the irradiation dose range from 0 to 5 kGy.

  1. there are many uses of jargon words, try to limit it.

Response: Thanks for your valuable suggestions. We have checked and revised the full text and marked it in red.

  1. Methods are the heart of the article,,,, try to detail every thing you used.. for example, in detailes explain how did you irradiate the fiber, fiber extraction, characterization and so on.

Response: Thanks for your valuable suggestions. All methods in the text have been described in detail and marked in red.

  1. why you did not identify the fibers structure?

Response: Thanks for your valuable suggestions. We determined that the structural changes of pea fibers were determined by measuring the changes in the content of main components, particle size and specific surface area of pea fibers, as well as by SEM, Fourier transform infrared spectroscopy and X-ray diffraction analysis.

  1. Most importantly, the underlying mechanism of the effect of the irradiation on the functional properties of fibers is REALLY missing, try to elucidate it, otherwise...

Response: Thanks for your valuable suggestions. We have elucidated in detail the mechanisms underlying the effects of γ-irradiation on the functional properties of pea fibers in the Results and Discussion section. We have marked the modified content in the text in red.

For example, when analyzing the content changes of the main components, we performed a cause analysis of changes in cellulose components from γ-irradiation and added citations. When analyzing the oil holding capacity, we elucidate that the high oil holding capacity of dietary fiber can reduce the absorption of lipids in the intestinal tract (Elleuch et al., 2011). When the γ-irradiation dose was 2 kGy, the oil holding capacity of pea dregs dietary fiber was the highest. This may be because of the γ-irradiation treatment can increase the porosity and specific surface area of pea fibers by destroying the microstructure. This is helpful for powering more oil droplets to contact and embed in pea fibers, thereby improving the oil holding capacity of pea dietary fibers (Al-Sheraji et al., 2012).

In the analysis of swelling and water holding capacity, we detailed the influence mechanism. With the increase of γ-irradiation dose, the swelling capacity and water holding capacity increased first and then decreased. And when the irradiation dose is 2 kGy, the swelling capacity and water holding capacity are the highest. It shows that the appropriate irradiation dose can significantly improve the swelling capacity and water-holding capacity of dietary fiber. Alam who reported in his study that particle size can affect the swelling capacity of dietary fiber, and the smaller the particle size, the higher the swelling capacity of dietary fiber (Alam et al., 2014). This is consistent with the results in 3.1.2. Besides, the moderate irradiation could break some fibers, thereby loosening the dense network structure of dietary fibers and increasing the specific surface area. This is also one of the reasons for the increased swelling capacity of dietary fiber. Irradiation treatment can break the glycosidic bond of dietary fiber, and the microstructure of fiber has obvious pores and honeycomb structure at 2 kGy. This provides more space for the storage of water molecules, thereby enhancing the water-holding capacity of the pea fibers(Xia et al., 2018; Fan et al., 2020).However, when the irradiation dose was too high, the swelling and holding capacity of pea fiber decreased significantly (P<0.05). This may be because the swelling capacity is related to the crystallinity.

When analyzing the nitrite adsorption of pea fiber, we elaborated the adsorption capacity of pea fiber samples for nitrite ions under simulated gastric environment (pH=2) and intestinal environment (pH=7). When pH=2, the nitrite adsorption capacity was much higher than that of pH=7. This may be because under acidic conditions, NO2- reacts with H+ to produce HNO2 and then forms nitrogen oxides, including the strong electron affinity compound N2O3, which can combine with the negatively charged oxygen atoms of phenolic acid groups in dietary fiber and cause adsorption (Lyu et al., 2021). Another explanation is that the active groups such as uronic acid, amino acid, especially phenolic acid contained in the structure of dietary fiber have a strong adsorption effect on nitroso groups under acidic conditions, that is why the nitrite ion adsorption capacity of pea fiber sample at pH = 2 is significantly higher than that at pH = 7 (Chu et al., 2019). With the increase of irradiation dose, the nitrite adsorption capacity of pea fiber increases first and then decreases at two pH values. When the irradiation dose is 2 kGy, the nitrite adsorption capacity of pea fiber was the highest. The γ-irradiation changed the structure of pea fibers, the surface appeared loose and porous, and the specific surface area became larger, which exposed more adsorption sites for nitrite ions to a certain extent. The capillary effect formed by these pores accelerates the entry of nitrite ions into the interior of the fiber structure, so the adsorption capacity of pea fibers for nitrite ions after moderate irradiation treatment is improved in the gastrointestinal environment.

In the analysis of cholesterol adsorption of pea fiber, we added that the γ-irradiation increases the specific surface area of the insoluble dietary fiber of pea dregs at irradiation doses below 2 kGy, and changes the structure of hemicellulose and lignin to enhance the capillary action, thereby increasing the adsorption capacity of cholesterol (Chen et al., 2014; Chu et al., 2019; Gan et al., 2020).

When analyzing the glucose adsorption of pea fiber, we describe the underlying mechanisms. The γ-irradiation could improve the glucose adsorption capacity of pea fibers, possibly due to the increase of specific surface area and porosity caused by moderate irradiation, making it easier for glucose to enter the interior of the fiber and bind more tightly to the interior of the dietary fiber. Studies have shown that the increased hydration of dietary fiber makes it easier for glucose to bond to the fiber's surface (Zhang et al., 2015). Furthermore, the adsorption capacity was positively correlated with glucose concentration. This may be due to the increased contact probability between glucose and fiber, which improves the adsorption capacity of pea fiber.

  1. The FTIR figure showed nothing, peak fit these results or provide much better one!! The same with XRD!

Response: We have marked the peak results in the FTIR figure and XRD figure, and elucidated the underlying mechanism and cited references in the results and analysis. Among them, the analysis results of Fourier transform infrared spectroscopy analyzed the changes of functional groups and main chemical bonds through the changes of peaks. The changes of γ-radiation on pea fibers crystallinity was analyzed by X-ray diffraction. The modified text has been highlighted in red.

Reviewer 4 Report

Review of manuscript foods-1680612 Effects of γ- irradiation on structure and functional properties of Pea fiber

 Manuscript deals with important topic and fits with topic to Foods journal. There are not great problems with presenting methods and results but I have to mention some systemic mistakes that can be easy removed.

  1. References in text do no correspond to guidance for authors: numbers in brackets such as [22] instead first author family name and year.
  2. Indexes in chemical substances and parameters of equations in text are at the same level as letters: e.g. at line 155 there is NaNO2. Number 2 should be lower index.
  3. Equations on lines 112, 122, 132, 135, 152, 166, 182 are not numbered. Referring on them is not easy.
  4. pH on lines 325 a 326 is written as “PH”.
  5. General remark: number of % should have gap between number and symbol %. E.g. “1.34 %” means that there is the content of some component 1.34 percent. If there is no gap between number and symbol % such as “6%” it can be interpreted as e.g. there is Natrium chloride water solution concentrate sixth percentual.

Author Response

Dear Editors and Reviewers:

Thank you for your letter and for the reviewers’ comments concerning our manuscript entitled “Effects of γ- irradiation on structure and functional properties of Pea fiber” (ID: foods-1680612). Those comments are all valuable and very helpful for revising and improving our paper, as well as the important guiding significance to our researches. We have studied comments carefully and have made correction which we hope meet with approval. Revised portion are marked in red in the paper. We hope that the revision is acceptable and look forward to hearing from you soon. The main corrections in the paper and the responds to the reviewer’s comments are as following:

Reviewer #4:

- Manuscript deals with important topic and fits with topic to Foods journal. There are not great problems with presenting methods and results but I have to mention some systemic mistakes that can be easy removed.

  1. References in text do no correspond to guidance for authors: numbers in brackets such as [22] instead first author family name and year.

Response: Thanks for your valuable suggestions. We have modified the reference style and the text and marked in red.

Corrected cited References:

Chen, J., Zhao, Q., Wang, L., Zha, S., Zhang, L., & Zhao, B. Physicochemical and functional properties of dietary fiber from maca (Lepidium meyenii Walp.) liquor residue. Carbohydrate Polymers. 2015, 132, 509-512.

Fan, X., Chang, H., Lin, Y., Zhao, X., Zhang, A., Li, S., Feng, Z., & Chen, X. Effects of ultrasound-assisted enzyme hydrolysis on the microstructure and physicochemical properties of okara fibers. Ultrasonics Sonochemistry. 2020, 69, 105247.

Lyu, B., Wang, H., Swallah, M. S., Fu, H., Shen, Y., Guo, Z., Feng, Z., & Jiang, L. Structure, properties and potential bioactivities of high-purity insoluble fibre from soybean dregs (Okara). Food Chemistry. 2021, 364, 130402.

Wu, C., Teng, F., McClements, D. J., Zhang, S., Li, Y., & Wang, Z. Effect of cavitation jet processing on the physicochemical properties and structural characteristics of okara dietary fiber. Food Research International. 2020, 134, 109251.

  1. Indexes in chemical substances and parameters of equations in text are at the same level as letters: e.g. at line 155 there is NaNO2. Number 2 should be lower index.

Response: Thanks for your valuable suggestions. We checked all Indexes in chemical substances and parameters of equations in the manuscript. We have modified it and marked in red.

  1. Equations on lines 112, 122, 132, 135, 152, 166, 182 are not numbered. Referring on them is not easy.

Response: Thanks for your valuable suggestions. We numbered all the formulas in the manuscript. We have modified it and marked in red.

  1. pH on lines 325 a 326 is written as “PH”.

Response: Thanks for your valuable suggestions. We checked the text and charts of the manuscript. We have modified it and marked in red.

  1. General remark: number of % should have gap between number and symbol %. E.g. “1.34 %” means that there is the content of some component 1.34 percent. If there is no gap between number and symbol % such as “6%” it can be interpreted as e.g. there is Natrium chloride water solution concentrate sixth percentual.

Response: Thanks for your valuable suggestions. We have modified it and marked in red.

Round 2

Reviewer 2 Report

The revised version has been notably improved. The research purpose and significance have been added in the Introduction section. All the methods in this paper are nowdescribed in detail. The underlying mechanism of the effect of γ-irradiation on the structural and functional properties of pea fiber has also been re-evaluated in the Results and Discussion section and a citation was added. Therefore, I recommend accepting this revised version.

Author Response

Dear Editors and Reviewers:

Thank you for your letter and for the reviewers’ comments concerning our manuscript entitled “Effects of γ- irradiation on structure and functional properties of Pea fiber” (ID: foods-1680612). Those comments are all valuable and very helpful for revising and improving our paper, as well as the important guiding significance to our researches. After your revision, the level of our paper has been greatly improved. We thank you for your dedication.

Reviewer 3 Report

It could be accepted in its current form.

Author Response

(The authors gave the same response as above.)
